# SCoT: Self-Correction at Test-time for Image Generation

## Abstract

Test-time scaling has emerged as an effective strategy to enhance image generation quality by repeatedly generating multiple images and selecting optimal outputs. However, such best-of-N schemes essentially rely on blind resampling with different random seeds, lacking the ability to incrementally refine errors based on previously correct generations. Some improved approaches rely on external verifiers to identify textual errors and feed them back to the model for refinement. However, they do not support targeted modifications with image consistency, and introduce further computational overhead. In this work, to address these limitations, we propose *Self-Correction at Test-time* (SCoT), a novel framework that equips generative models with internal self-assessment and targeted revision capabilities. Specifically, SCoT is trained to preserve the correctly generated regions while autonomously modifying only erroneous parts, eliminating the need for external guidance. This self-reflective mechanism enhances visual consistency, and unlocks the model's potential capacity for prompt-guided correction. SCoT improves over the baseline by up to 0.25, substantially surpassing prior methods, providing a more reliable, efficient, and user-aligned approach to high-quality image generation.

## 1 Introduction

Humans have a natural drive for self-expression. When inspired, we mentally construct scenes and aspire to render them vividly so as to communicate and interact with others. Generative models, by exploiting their learned knowledge of data manifolds, provide a means to synthesize high-quality images according to user specifications. However, the generated results are not always accurate; errors in object placement, attributes, or other details may fail to meet user expectations. To address this issue, users typically resort to generating multiple samples with different random seeds, or iteratively adjusting the initial prompts they intend to express, in the hope that the model will eventually produce the desired result. As this process is largely governed by randomness, users may, after numerous unguided attempts, still be unable to achieve their intended outcomes and are thus compelled to give up. This inherent inefficiency undermines reliability and introduces a barrier to the broader deployment and acceptance of such technology.

How can we make the outputs of generative models more reliable and more likely to meet user expectations? The core challenge lies in the model's limited ability to understand its own generations. A key question, therefore, is how to enable the model to develop self-understanding and to revise its outputs in a purposeful and directed manner.

Recent advances in image generation models, such as diffusion and transformer frameworks (Peebles & Xie, 2023; Esser et al., 2024a; Labs, 2024; Esser et al., 2024b), have demonstrated remarkable capabilities in producing high-quality visual content. However, the quality of generated images is often sensitive to the choice of random seeds and inference trajectories. To mitigate this, existing test-time scaling methods typically rely on multiple sampling attempts (Xie et al., 2025) or external verifiers to recognize errors (Li et al., 2025). While effective to some extent, these approaches suffer from several limitations: they often require blind exploration of new seeds, frequently alter the overall image layout, disrupt visual consistency, and introduce additional computational overhead.

In fact, these limitations fundamentally come from the inability of existing methods to autonomously identify and correct errors. Current approaches lack intrinsic mechanisms enabling models to inter-

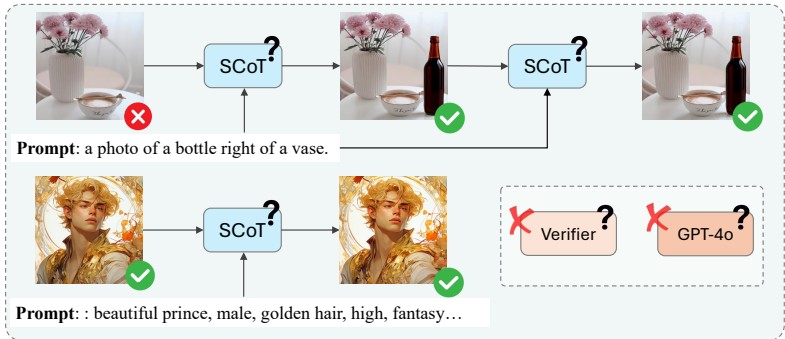

Figure 1: Overview of our method. Our method embeds judgment and reasoning into the generative model, thereby activating its inherent capacity for understanding and self-correction. After training, our method demonstrates robust generalization to real images and diverse prompt domains.

nally evaluate their outputs and selectively modify specific image regions. Instead of relying on external signals or brute-force resampling, we argue that a more promising direction is to stimulate the model's own reflective reasoning capacity. A generative model inherently encodes rich knowledge of both prompts and images, yet this capacity has not been systematically exploited as a source of self-corrective potential.

Motivated by this, we propose *Self-Correction at Test-time* (SCoT), a novel framework that equips generative models with an internal capacity for self-assessment and selective revision during inference. Unlike conventional approaches that indiscriminately resample entire images, SCoT identifies erroneous regions and precisely modifies them, preserving already accurate content. This targeted self-correction removes reliance on external verifiers, substantially enhancing visual consistency and computational efficiency, while effectively unlocking the model's inherent reflective capabilities.

Compared to prior methods, SCoT offers several distinctive advantages. First, it maintains visual and structural consistency by retaining correct regions across iterations. Second, it significantly reduces unnecessary sampling and computation by focusing modifications only where needed. Third, it provides a novel perspective on exploring a model's internal understanding and generative potential, highlighting its ability to self-correct and refine outputs autonomously.

Overall, SCoT introduces a new paradigm for test-time inference in image generation, emphasizing self-reflective, localized, and visually consistent modifications. On the GenEval benchmark, our method improves over the baseline by up to 0.25, substantially surpassing all other approaches. For tasks that require deeper image understanding, such as relative position and attribution binding, our method delivers much greater improvements.

## 2 RELATED WORK

**Generative models.** Research on generative models spans multiple paradigms. GANs (Brock, 2018; Goodfellow et al., 2014; Karras, 2019) pioneered high-fidelity image synthesis but suffer from unstable training. VAEs (Kingma, 2013) improve stability but often generate blurry outputs. Autoregressive models (Tian et al., 2024; Sun et al., 2024) capture rich dependencies by sequentially predicting tokens, though their autoregressive nature incurs prohibitive costs for high-resolution images. Diffusion models (Ho et al., 2020; Sohl-Dickstein et al., 2015; Song et al., 2020b) have recently become the leading framework, offering both stability and high quality. A number of extensions (Song et al., 2020a; Nichol et al., 2021; Ye et al., 2023; Zhang et al., 2023) further enhance them by improving controllability and sampling efficiency.

**Image modification.** Image modification has been approached using both training-free and instruction-based methods. Traditional methods like SDEdit (Meng et al., 2021) can directly adjust images conditioned on the input prompt. In addition, there exist other training-free methods such as Prompt-to-Prompt (Hertz et al., 2022), MasaCtrl (Cao et al., 2023) and Plug-and-Play (Tumanyan et al., 2023). They manipulate images by carefully controlling internal features during different generation forward, but are cumbersome and not fully end-to-end. Most instruction-based models

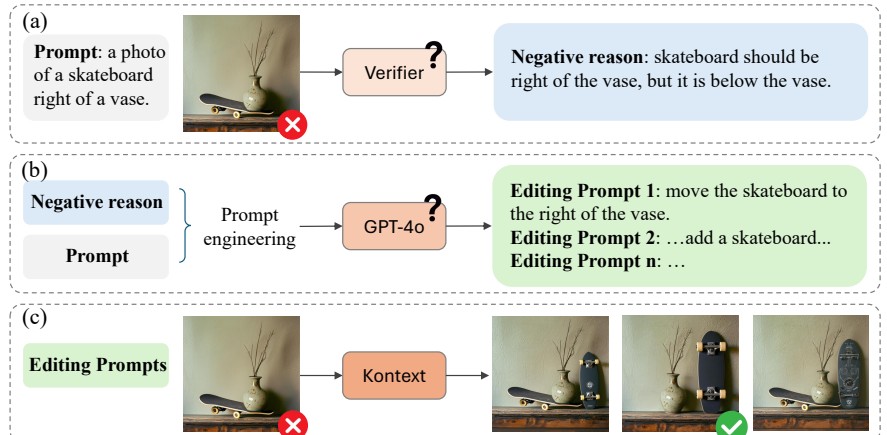

Figure 2: Constructing positive–negative pairs. (a) We first collect negative images and corresponding negative reasons from object detectors. (b) Then we analyze negative examples and use GPT-4o to generate editing prompts. (c) After that, FLUX.1-Kontext-dev model is employed to generate corresponding corrected images. Those images are further filtered.

(Brooks et al., 2023; Labs et al., 2025; Wu et al., 2025), are typically trained on large-scale datasets with editing instructions. Differently, our method directly exploits the model's ability to reason upon positive and negative cases conditioned on the original prompt, thereby enabling correction without any editing prompts, providing a feedforward end-to-end solution.

**Test-time scaling.** Prior works on test-time scaling typically rely on generating multiple candidates and reranking (Ma et al., 2025; Singhal et al., 2025; Xie et al., 2025), which improves image quality at the cost of computation and consistency. Another line of research explores iterative refinement (Li et al., 2025; Wu et al., 2024; Wang et al., 2024; Yu et al., 2023), where models or auxiliary systems progressively improve outputs, but such methods often require external verifiers or user guidance. Related efforts on image editing and consistency preservation (Brooks et al., 2023; Labs et al., 2025; Wu et al., 2025; Tian et al., 2024) can constrain modifications, yet they generally depend on conditioning signals or prompt engineering.

In contrast, our proposed SCoT framework requires neither an external verifier to identify textual errors nor additional editing prompts. It enables the model to self-assess and selectively modify only the necessary regions. After training, our method demonstrates inference, maintaining global consistency while improving quality. This highlights a new direction for leveraging models' intrinsic self-judgment in scaling generative performance.

## 3 METHOD

We aim to elicit the intrinsic ability of the model to self-evaluate and correct. Inspired by Brooks et al. (2023), we formulate this as a supervised learning task involving two main stages: (1) constructing positive–negative training pairs by pairing mismatched prompts with images before and after correction (Section 3.1, Fig. 2); and (2) training a self-correcting model on this generated dataset without any external error signals (Section 3.2, Fig. 1). Although trained solely on synthetic image pairs, our method demonstrates strong generalization capabilities, effectively transferring to real images and diverse prompt domains. See Fig. 1 for an overview of our method.

### 3.1 GENERATING A TRAINING DATASET

To promote self-correction under test-time scaling, it is essential to construct high-quality training data. A critical component is the integration of incorrect images as negatives, which serve to establish positive–negative pairs and guide the model toward a deeper understanding of what constitutes good and poor generations, as shown in Fig 2.

### 3.1.1 CONSTRUCTING POSITIVE–NEGATIVE PAIRS.

We leveraged the 78.5k image–feedback pairs released in Reflect-DiT (Li et al., 2025) to construct our training dataset. These prompts are constructed from GenEval templates, with those appearing in the test set filtered out. In detail, there are approximately 3.9k prompts in Reflect-DiT, and each is associated with 20 generated images that include both positive and negative examples, and corresponding feedback is obtained from object detectors (Ghosh et al., 2023).

Based on this, we conduct an analysis of negative examples, empirically select the most effective editing prompt for each error type, and construct the corresponding instructions. Next, GPT-4o (Hurst et al., 2024) is used under these instructions to generate editing prompts for each negative image, conditioned on the image's original prompt and object-detector feedback. Using this editing prompt, we adopt FLUX.1-Kontext-dev (Labs et al., 2025) to generate the corresponding corrected image. We then select the images that are successfully corrected and apply image quality metrics for further filtering, yielding 14.8k aligned positive–negative pairs. The process of rectifying prompt-violating generated images is presented in Fig. 2.

### 3.1.2 ANALYSIS ON THE GENERATION PIPELINE

Note that this process is highly labor-intensive, and it underscores the practical challenges users face when using generative models. Without **built-in self-evaluation** and **correction**, the model leaves users to assess whether generated images meet their expectations, identify specific inconsistencies, and craft appropriate editing prompts, followed by iteratively refining and testing these editing prompts until the desired result is achieved.

We argue that the root cause lies in the lack of prompt–image reasoning capability within the generative model, leaving generation and understanding disjointed rather than mutually reinforcing. To tackle this problem, we explore activating the model's intrinsic evaluation and self-correction capabilities. We provide a detailed description of our model in the following Section 3.2. Moreover, in order to help the model better distinguish between positive and negative examples, we augmented the positive–negative pairs with an additional 30% of purely positive pairs.

As depicted in Fig. 1, our model exhibits a new level of capability in understanding images and prompts. It requires neither an external verifier to identify errors nor additional editing prompts. Given an incorrect image as a condition, SCoT can comprehend and rectify mismatches with the prompt by itself, and then output the corrected image directly. For images that already conform to the prompt, it preserves all details and outputs them almost unchanged. Despite being trained on generated image pairs, our method generalizes well to real images and diverse prompt domains.

### 3.2 SELF-CORRECTION MODEL

We use our generated data to train a model that is capable of Self-Correction at Test time (SCoT) without additional feedback. Following Reflect-DiT (Li et al., 2025), we utilize SANA-1.0-1.6B (Xie et al., 2024) as the base model due to its relatively small size, low inference cost, and fast sampling speed, making it well-suited for inference-time scaling that involves generating many samples per prompt.

### 3.2.1 FULLY PRESERVE IMAGE INFORMATION

Flow-based methods (Lipman et al., 2022; Liu et al., 2022; Xie et al., 2024) regard the denoising process as probability density flow, modeling the vector field $u_t(x)$ with a neural network:

$$\mathcal{L}_{FM} := \mathbb{E}_{t,p_t(x)}\left[\|v_t(x) - u_t(x)\|_2^2\right] \tag{1}$$

where $p_t(x)$ represents the probability density path, $x \sim p_t(x)$ and $t \sim \mathbb{U}[0,1]$. In text-to-image generation scenarios, the model with parameters $\theta$ receives the time step $t$, text prompt $C_T$, and noisy image features $X$ as input, and outputs the corresponding velocity $v_\theta(\cdot)$ at that moment.

Subsequently, many methods introduced image conditions $C_I$ into generative models, aiming to provide finer-grained spatial guidance (Zhang et al., 2023; Ye et al., 2023; Brooks et al., 2023; Tan et al., 2024; 2025; Labs et al., 2025) or to enable the model to perform test-time reflection based

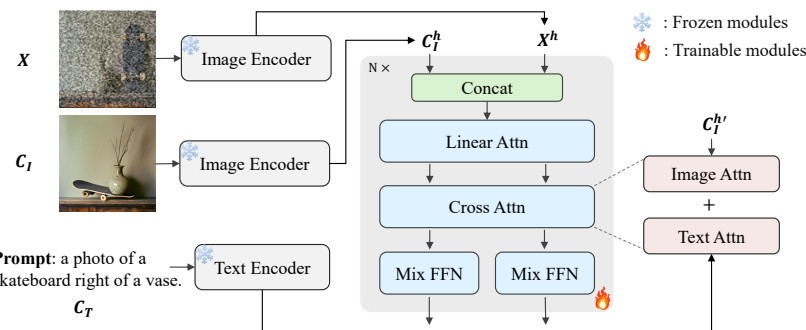

Figure 3: Architecture of SCoT. In our model, both the image condition $C_I$ and the noisy image input $X$ are processed by the VAE encoder to fully preserve image information. To align the features of the image condition with the noisy image input, both are passed through the Linear-DiT blocks. We embed information-sharing mechanisms within both linear and cross attention.

on previously generated images (Li et al., 2025). Among these, the most common method for extracting information from the image condition $C_I$ is to use an extra pre-trained image encoder, for instance, the CLIP (Radford et al., 2021) or SigLIP (Zhai et al., 2023) image encoder, to obtain image representations. However, due to the limitations of the pre-trained image encoder and the discrepancy between the encoder's embedding space and the feature space of noisy image inputs, this approach results in a significant loss of image condition information, as further illustrated in Section. 4.2.

Since our approach requires the model to reason over and partially rectify the input image while ensuring consistency across other regions, it is crucial to maximize the preservation of image condition information. To this end, we use a fully consistent approach to extract and embed both the image condition and the noisy input, as shown in Fig. 3. In our model, both the image condition $C_I$ and the noisy image input $X$ are processed by the VAE encoder, mapped into the latent space, and represented as $C_I^h \in \mathbb{R}^{N \times d}$ and $X^h \in \mathbb{R}^{N \times d}$. Similarly, the original text prompt is fed through the text model to extract its feature embedding $C_T^h \in \mathbb{R}^{M \times d}$. Here, $d$ denotes the embedding dimension, while $N$ and $M$ represent the number of image and text tokens, respectively.

### 3.2.2 INCORPORATING IMAGE CONDITION

We adjusted the model architecture to better incorporate image condition, as depicted in Fig. 3. The original SANA model is composed of stacked Linear-DiT blocks, each including linear-attention, cross-attention, and a feed-forward network. To align the features of the image condition with the noisy image input, both are passed through the Linear-DiT blocks.

And for more effective interaction with the image condition, we embed information-sharing mechanisms within both linear and cross attention. Before entering the linear attention, the two features $C_I^h$ and $X^h$ are concatenated, enabling interaction through self-attention. Subsequently, in the cross-attention module, inspired by Ye et al. (2023), we introduce an additional interaction branch for image condition, which runs in parallel with the original text condition branch, as shown in Fig. 3.

More specifically, hidden image tokens are projected into queries $Q$ via the multi-head attention mechanism. In the standard text condition branch, text embeddings are similarly mapped to key $K_T$ and value $V_T$. To incorporate image conditions, we further introduce a newly designed projection module, parameterized by $W_k$ and $W_v$, to map image condition tokens into a space that facilitates conditional understanding and propagation. After that, we can get projected image conditions $K_I$ and $V_I$. The outputs of both cross-attention branches are then aggregated to form the final cross-attention result for the image, as:

$$\text{MHA}(X^h, C_T^h, C_I^h) = \text{Softmax}(\frac{QK_T^\top}{\sqrt{d}})V_T + \text{Softmax}(\frac{QK_I^\top}{\sqrt{d}})V_I \tag{2}$$

Moreover, to further enhance reflective integration between image and text conditions, the image condition tokens are allowed to interact with text tokens prior to entering the dual-branch cross-

Table 1: Results on the GenEval benchmark. Our method achieves the highest overall score 0.91, surpassing all other approaches. Relative to Reflect-DiT and SFT, our approach exhibits a substantially higher performance. † models are fine-tuned on the same training data. ‡ SANA-1.5 results are reported from the original paper.

| Generator | Params | Overall | Single | Two | Counting | Color | Position | Attribution |
|---|---|---|---|---|---|---|---|---|
| SDXL(Podell et al., 2023) | 2.6B | 0.55 | 0.98 | 0.74 | 0.39 | 0.85 | 0.15 | 0.23 |
| DALLE 3(Betker et al., 2023) | - | 0.67 | 0.96 | 0.87 | 0.47 | 0.83 | 0.43 | 0.45 |
| SD3(Esser et al., 2024a) | 8B | 0.74 | 0.99 | 0.94 | 0.72 | 0.89 | 0.33 | 0.60 |
| Flux.1-Dev(Labs, 2024) | 12B | 0.68 | 0.99 | 0.85 | 0.74 | 0.79 | 0.21 | 0.48 |
| Playground v3(Liu et al., 2024) | - | 0.76 | 0.99 | 0.95 | 0.72 | 0.82 | 0.50 | 0.54 |
| SANA-1.5-4.8B Pre (Xie et al., 2025) ‡ | 4.8B | 0.72 | 0.99 | 0.85 | 0.77 | 0.87 | 0.34 | 0.54 |
| + Best-of-2048 ‡ | 4.8B | 0.80 | 0.99 | 0.88 | 0.77 | 0.90 | 0.47 | 0.74 |
| SANA-1.0-1.6B (Xie et al., 2024) | 1.6B | 0.66 | 0.99 | 0.77 | 0.62 | 0.88 | 0.21 | 0.47 |
| + SFT (Best-of-20) † | 1.6B | 0.87 | 1.00 | 0.98 | 0.83 | 0.91 | 0.81 | 0.70 |
| + Reflect-DiT (N=20) | 1.6B + 0.1B | 0.81 | 0.98 | 0.96 | 0.80 | 0.88 | 0.66 | 0.60 |
| + Reflect-DiT (N=20) † | 1.6B + 0.1B | 0.78 | 0.99 | 0.91 | 0.74 | 0.88 | 0.66 | 0.55 |
| + Ours (Best-of-20) | 1.6B + 0.2B | **0.91** | **1.00** | **1.00** | **0.86** | **0.95** | **0.90** | **0.77** |
| (△ vs Baseline) | - | +0.25 | +0.01 | +0.23 | +0.24 | +0.07 | +0.69 | +0.30 |

attention mapping. Additionally, since prior studies (Islam et al., 2020; Xie et al., 2021), both theoretical and empirical, suggest that applying $3 \times 3$ convolutions with zero padding implicitly embeds positional information, SANA removes Positional Embedding and utilizes the positional bias inherent in convolution to convey position information. To maintain this positional inductive bias within the Mix-FFN, we do not incorporate interaction inside the convolutional layer.

As illustrated in ControlNet (Zhang et al., 2023), zero-initialization allows for the gradual injection of image condition information. Following this idea, we initialize $W_k$ and $W_v$ to zero. Our model is initialized with the released weights of SANA-1.0-1.6B, thereby preserving its native generative capacity. As shown in Fig. 3, parameters of the image and text encoders remain frozen, while other modules are updated during fine-tuning. We adopt the SANA training objective, with a 0.1 probability of dropping the image condition during training.

## 4 EXPERIMENTS

### 4.1 SETUP

**Implementation details.** We build our method upon SANA-1.0-1.6B (Xie et al., 2024), which is a high-efficiency flow-based model for image generation. We train our model with a batch size of 16 and employ the Prodigy optimizer (Mishchenko & Defazio, 2024) with safeguard warmup and bias correction enabled, and a weight decay of 0.01. Experiments were executed on 4 NVIDIA H100 GPUs (80 GB each), with the model trained for 60,000 iterations, finishing in one day.

**Baselines.** Through self-correction on produced outputs, our model adapts naturally to test-time scaling, eliminating the need for an additional Vision Language Model (VLM) to provide textual feedback as used in prior work (Li et al., 2025). Our method establishes a novel paradigm for self-reflection in inference. Therefore, we first compare different approaches in the test-time scaling stage, such as Reflect-DiT (Li et al., 2025) and naive best-of-N sampling. Next, to assess our method's one-step self-correction under the original prompt, we test it against both classic and cutting-edge image-modifying models (Meng et al., 2021; Brooks et al., 2023; Labs et al., 2025; Wu et al., 2025), further highlighting its uniqueness and performance.

### 4.2 BASELINE COMPARISONS

In this section, we comprehensively evaluate the effectiveness of our proposed method by comparing it against multiple baseline approaches, conducting both quantitative and qualitative analyses within the test-time scaling setting.

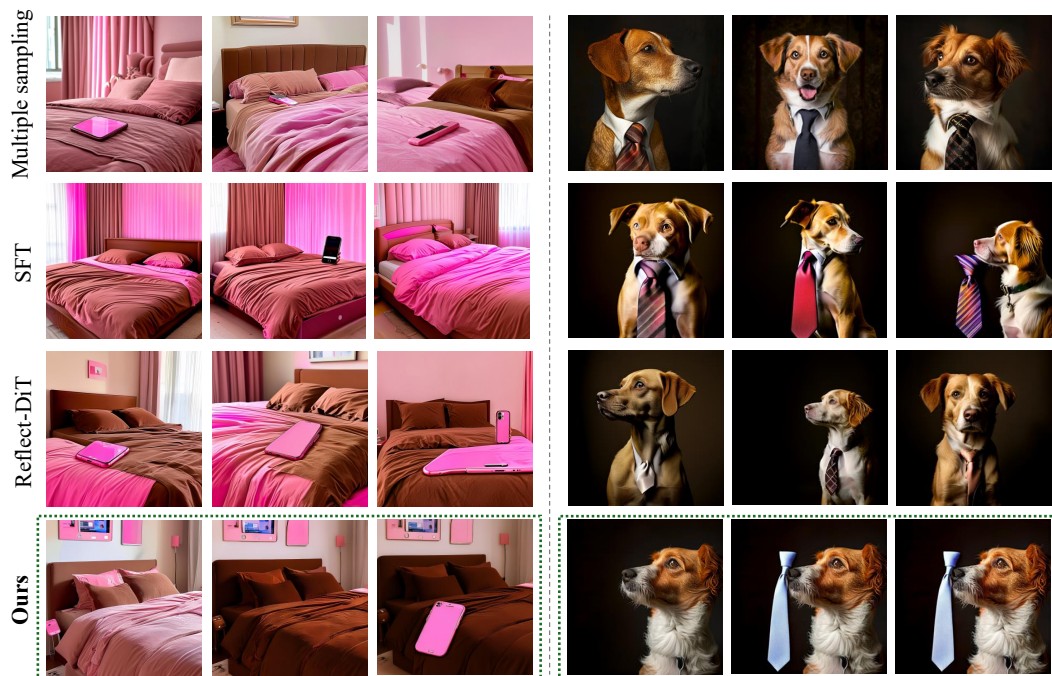

Prompt: a photo of a brown bed and a pink cell phone.     Prompt: a photo of a dog right of a tie.

Figure 4: Visualization results on the GenEval test set. Our method can leverage previously generated images as conditions to guide corrective attempts. Compared to Reflect-DiT, our approach achieves a deeper understanding of the image, selectively identifying regions requiring correction while preserving other areas, thereby enabling more purposeful and targeted improvements.

### 4.2.1 QUANTITATIVE COMPARISON

The results on the GenEval benchmark (Ghosh et al., 2023) are summarized in Tab. 1. For a fair evaluation, we retrain Reflect-DiT on our dataset, indicated by †. Additionally, the original SANA model is fine-tuned on the same data, and we compare our results with the best-of-N outcomes obtained after fine-tuning. For each randomly generated result from the base model, we perform a single step of self-correction. Moreover, to provide a rigorous comparison, we strictly limit the total number of generation runs in the best-of-N evaluation, i.e., a maximum of N pipeline executions. To be consistent with Reflect-DiT's evaluation, the metrics based on SANA-1.0-1.6B are computed using the single best output out of max 20 generation runs.

As shown in Tab. 1, our method achieves the highest overall score (0.91) under max 20 generation runs, surpassing all other approaches, including the SANA-1.5–4.8B variant, which has more than $3\times$ parameters and employs best-of-2048 sampling. And our method outperforms the baseline by as much as 0.25. Moreover, relative to Reflect-DiT, our approach exhibits a substantially higher performance (0.91 vs. 0.78). When compared with the best-of-20 approach after SFT, our method yields greater benefits with only half of the randomly generated images. More importantly, for tasks that require deeper image understanding, such as relative position and attribution binding, our method delivers substantially greater improvements. This demonstrates that our SCoT method can effectively activate the generative model's deep understanding of images. Accurate generation relies on such comprehension, and fostering their interaction leads to mutual reinforcement.

### 4.2.2 QUALITATIVE COMPARISON

Fig. 4 presents visual results of different methods on the GenEval test set. It can be observed that, whether before or after SFT, multiple-sampling results rely purely on random attempts. In contrast, self-reflection methods, such as Reflect-DiT and our method, can leverage previously generated images as conditions to guide corrective attempts.

Table 2: Correction results on the GenEval benchmark. We achieves the best one-step correction performance. Moreover, our method exhibits the highest CLIP-Image similarity with the original image, highlighting its ability to preserve fine details and structural consistency.

| Generator | Params | GenEval ↑ | CLIP-I ↑ |
|---|---|---|---|
| SANA-1.0-1.6B (Xie et al., 2024) | 1.6B | 0.66 | - |
| InstructPix2Pix (Brooks et al., 2023) | 0.9B | 0.53 | 0.91 |
| FLUX.1-Kontext-dev (Labs et al., 2025) | 12B | 0.72 | 0.95 |
| OmniGen2 (Wu et al., 2025) | 4B | 0.73 | 0.93 |
| + Ours (Self-Correction) | 1.6B + 0.2B | **0.76** | **0.96** |

However, Reflect-DiT cannot modify images continuously while preserving correct regions, such as adjusting the bed color while preserving scene layout. In contrast, our method can continuously alter specific regions while maintaining the scene structure, such as changing the bed color and adding a pink phone without affecting the room layout. Furthermore, in the second case, our approach exhibits prompt-guided self-reasoning capability, allowing the model to identify the absent tie on its own, without relying on an external VLM to indicate the error textually, as is required in Reflect-DiT.

Compared to Reflect-DiT, our approach achieves a deeper understanding of the image, selectively identifying regions requiring correction while preserving other areas, thereby enabling more purposeful and targeted improvements. More visualization results can be seen in the Appendix. A.

### 4.3 MORE RESULTS

Since our method also involves image modification, we compare it against representative image-editing approaches, including SDEdit (Meng et al., 2021), InstructPix2Pix (Brooks et al., 2023), OmniGen2 (Wu et al., 2025), and FLUX.1-Kontext-dev (Labs et al., 2025).

#### 4.3.1 QUANTITATIVE COMPARISON

Most existing image editing models are trained on large-scale datasets of editing instructions. Differently, our method directly exploits the model's ability to reason upon positive and negative cases against the given prompt, thereby enabling correction without any editing prompts. In Tab. 2, we compare results on the GenEval benchmark, where images generated by the SANA base model are refined 1 step conditioned on the original prompt.

As shown in Tab. 2, while recent image editing models are capable of partial corrections when conditioned solely on the image prompt, their corrections remain suboptimal. In contrast, our approach achieves the best one-step correction performance, substantially surpassing all baselines and reaching 0.76. Moreover, our method attains the highest CLIP-Image similarity with the original image, highlighting its ability to preserve fine details and structural consistency while leveraging prompt-based reasoning for targeted modifications.

#### 4.3.2 QUALITATIVE COMPARISON

Fig. 5 illustrates the visual comparison with other image editing baselines. Traditional methods like SDEdit can directly adjust images conditioned on the input prompt. We implement SDEdit based on SD1.5 (Rombach et al., 2022). It can be observed that both SDEdit and InstructPix2Pix exhibit limitations in adding missing objects or correcting the relative positions of objects. Moreover, these methods often lead to distortions in layout or inconsistencies in global style.

For the recent image editing model OmniGen2, we evaluate its performance under the original prompt. The results reveal that, when guided only by the raw prompt, OmniGen2 can sometimes fail to realize specific modifications (e.g., changing the color of an orange). And the fine details are not well preserved, such as modifications to the car's outer surface. In comparison, our method keeps both the main subject and surrounding context intact, leveraging the original prompt to perform autonomous, reflective corrections of areas inconsistent with the prompt, without requiring designed editing prompts. Additionally, while trained on synthetic data, it generalizes well to real images,

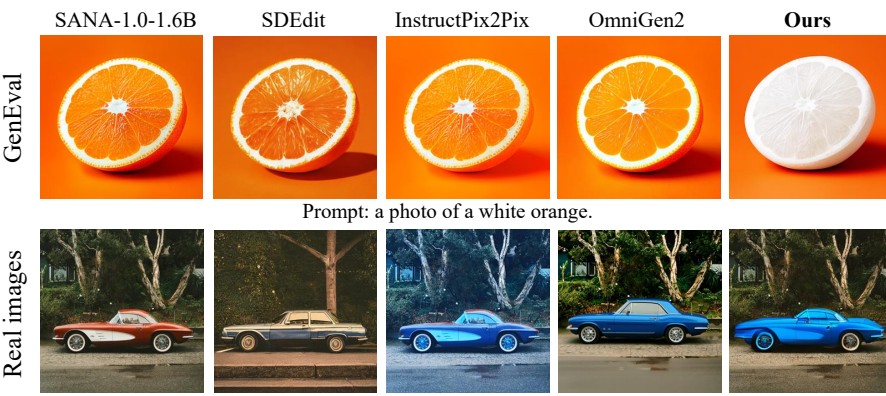

Figure 5: Visual comparison with other image editing baselines. Our method generalizes well to real images, highlighting its ability to learn transferable reflective reasoning skills.

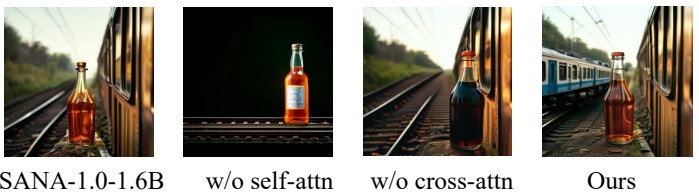

Figure 6: Ablation on interaction structure.

highlighting the model's ability to learn transferable reflective reasoning skills. More visualization results can be seen in the Appendix. A.

### 4.4 ABLATIONS

**Interaction structure.** We explored the interaction design of the model and found that whether interactions are missing in the self-attention or in the cross-attention layers, the model tends to struggle with maintaining the overall layout or the consistency of the main object, as shown in Fig. 6. Therefore, performing sufficient interactions in both is a more appropriate approach.

**Integration of negative examples.** We investigated the effectiveness of introducing negative examples. In Fig. 4, the SFT baseline is trained solely on positive examples. As shown, this limitation prevents the model from exhibiting self-correction capabilities. The comparison demonstrates that incorporating negative-positive pairs better stimulates the model to perform prompt-guided reflective reasoning and autonomous correction. We hypothesize that this is because introducing negative–positive pairs offers crucial contrastive signals, which provide important guidance for bridging image understanding and generation. Moreover, in Tab. 1, compared to best-of-20 sampling under SFT, our method achieves superior results with only half the number of randomly generated images, highlighting that fewer but more targeted interventions can outperform brute-force randomness. This not only improves efficiency but also offers users a more focused and satisfactory experience by reducing unnecessary trial-and-error.

## 5 CONCLUSION

We presented *Self-Correction at Test-time* (SCoT), a framework that enables generative models to autonomously assess and refine their outputs during inference. By leveraging inherent knowledge of prompts and images, SCoT activates self-reflective reasoning to selectively correct erroneous regions while preserving correct structures. Extensive experiments show that SCoT outperforms existing baselines in fidelity, structural consistency, and prompt-aligned corrections. Our approach highlights the latent self-correction potential of generative models and opens new avenues for more reliable, user-aligned, high-fidelity image synthesis.

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

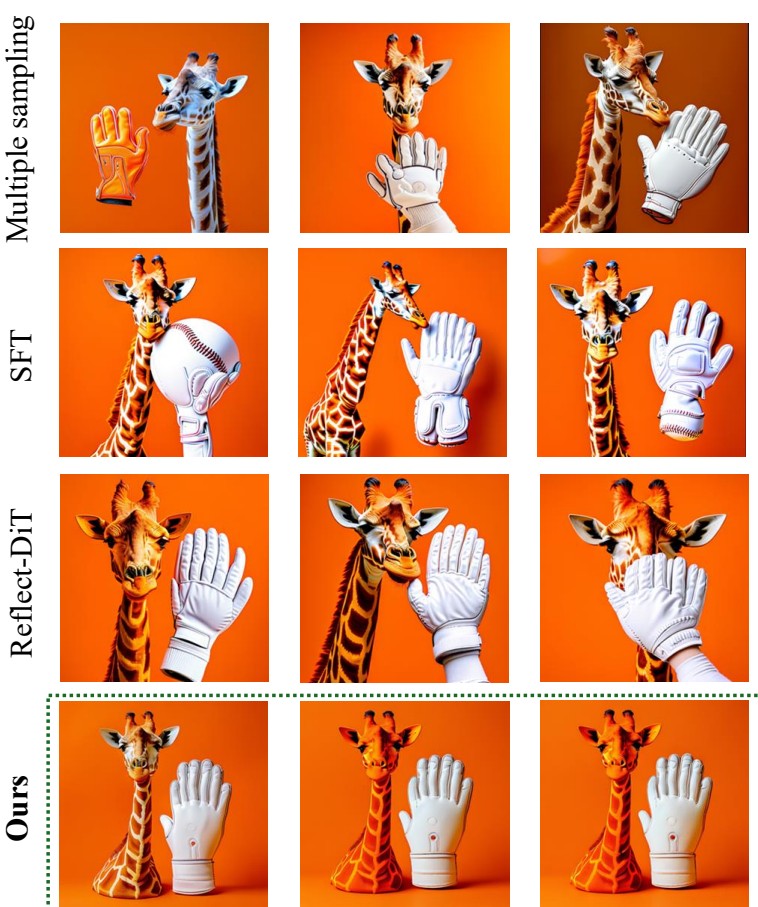

Prompt: a photo of an orange giraffe and a white baseball glove.

Figure 7: More visualization results on the GenEval test set.

## A    MORE RESULTS

Here, we provide additional visualization results of our model, including comparisons with test-time scaling and image editing models, as shown in Fig. 7 and Fig. 8. We observe that our model can adjust the giraffe's color without altering the gloves' appearance. In contrast to conventional image editing methods, it enables precise modifications to object binding and relative attributes, while also supporting the insertion of additional objects with specific properties. On real-world images, it demonstrates stronger detail preservation, for instance, retaining the chair behind the vase and maintaining the spatial arrangement of the baseball bat and the book.

## B    LLM USAGE

In this work, we use ChatGPT [1] to polish our sentences and check grammar. In our experiments, we also leverage the GPT-4o large model as a tool to generate editing prompts for the dataset.

---

[1]https://chatgpt.com/

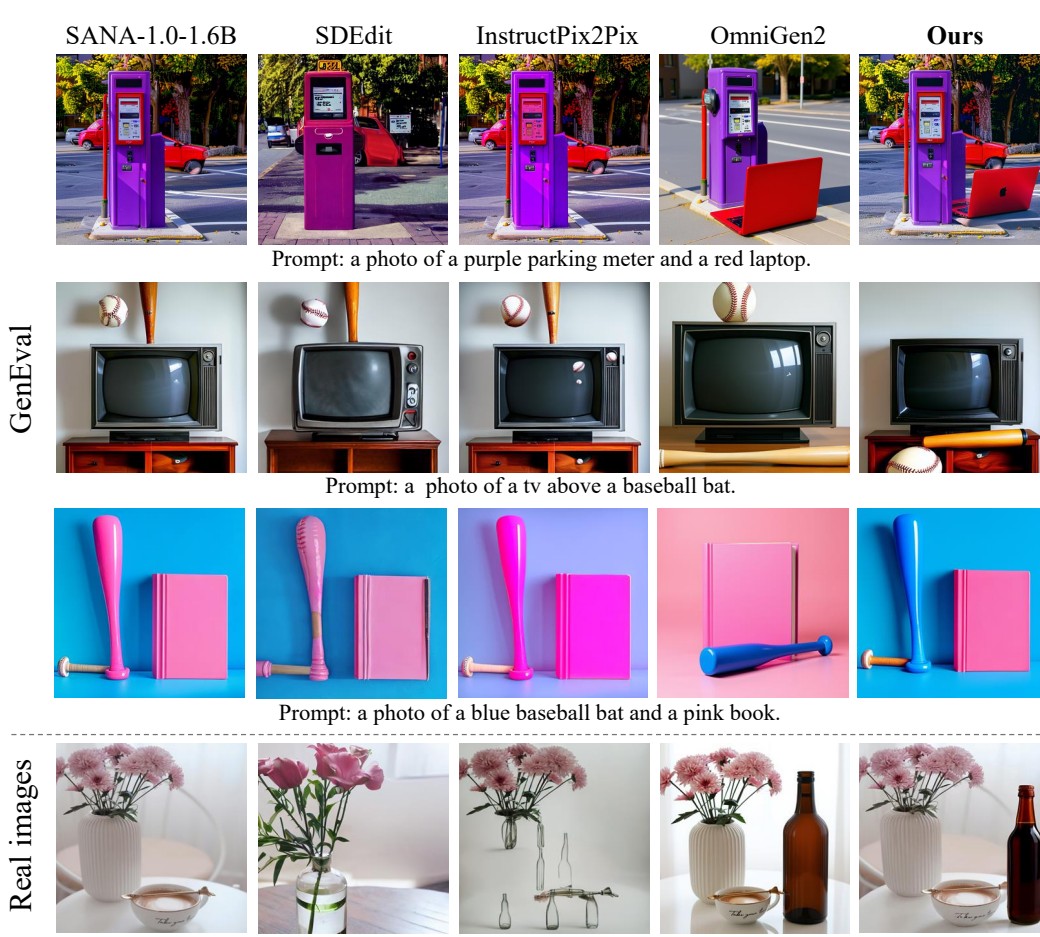

Figure 8: More visual comparison with other image editing baselines.

