# OpenReview forum: "SCoT: Self-Correction at Test-time for Image Generation"
_ICLR.cc/2026/Conference — ICLR 2026 Conference Withdrawn Submission_

### Official Review · Reviewer_cF4M · 2025-10-31

**Soundness:** 3
**Presentation:** 3
**Contribution:** 2
**Rating:** 4
**Confidence:** 5

**Summary:**

This paper proposes the Self-Correction at Test-time (SCoT) framework, which aims to address the issues of blind resampling, external dependence, and consistency loss in the "test-time scaling" methods for image generation. By constructing positive-negative sample pairs to train the model, this framework is based on the base model SANA-1.0-1.6B. It employs a VAE encoder to preserve image information and adopts a dual-branch cross-attention mechanism to enable image-text interaction, thereby endowing the model with the ability to perform local error correction without external guidance. Experimental results show that SCoT achieves a maximum improvement of 0.25 over the baseline on the GenEval benchmark, with an overall score of 0.91, outperforming methods such as  SFT and Reflect-DiT.

**Strengths:**

The paper presents comprehensive experiments, with clear advantages in subtasks such as relative position and attribute binding, which validates the effectiveness of the proposed method. Meanwhile, the paper’s presentation is relatively thorough, and aspects including the model architecture and data construction are relatively clear.

**Weaknesses:**

1) The authors’ core claim regarding innovation hinges on the assertion that "existing works such as Reflect-DiT rely on external verifiers or user guidance," while framing their proposed method as a breakthrough in achieving autonomous correction without external support. However, the existence of prior works like [1] (which demonstrates self-check capabilities in models such as BAGEL) does introduce potential challenges to this innovation claim—the extent of the impact depends on whether the authors explicitly clarify the essential differences between their method and BAGEL-like self-check mechanisms, what unique pain points does your method solve that BAGEL cannot? For example, does it outperform BAGEL in localized error precision, computational efficiency, or generalization to complex image generation scenarios (e.g., high-resolution images or ambiguous prompts)?
[1] Uni-cot: Towards Unified Chain-of-Thought Reasoning Across Text and Vision
2) I have noticed that the paper Reflect-DiT also uses the example "a photo of a dog right of a tie"; however, Reflect-DiT can achieve correct generation for this case, which is inconsistent with the demonstration presented in Figure 4 of this paper. Could this discrepancy be attributed to certain inconsistencies in experimental settings?
3)   In the ablation experiments, I would like to further see the authors' insights on self-attention and cross-attention. For instance, I observe that w/o cross-attn, the overall composition of the generated images does not undergo significant modifications; whereas w/o self-attn, there are substantial differences in the composition. Is this a universal phenomenon? Could the specific roles of these two types of attention in the proposed framework be further analyzed?

**Questions:**

Please see the weakness above.

---

### Official Review · Reviewer_qXdx · 2025-11-01

**Soundness:** 3
**Presentation:** 4
**Contribution:** 3
**Rating:** 4
**Confidence:** 4

**Summary:**

This paper introduces SCoT (Self-Correction at Test-time), a novel framework for improving text-to-image generation reliability.
The core idea is to fine-tune a model to intrinsically identify and correct its own errors, moving beyond blind resampling or reliance on external verifiers.
The method constructs a synthetic dataset of {prompt, incorrect_image, correct_image} triplets and trains a model to perform targeted, single-step corrections.
Architecturally, it uses a shared VAE encoder for both the input and conditional image to prevent domain gaps and employs a dual-branch cross-attention mechanism to integrate text and image guidance.
The method demonstrates significant performance gains on the GenEval benchmark, particularly on challenging compositional tasks like object positioning and attribute binding, while showing superior visual consistency compared to other image editing baselines.

**Strengths:**

- Novel and Relevant Problem Formulation:
The paper addresses a critical challenge in generative AI—reliability—by proposing an elegant self-correction mechanism that is learned directly into the model, avoiding complex multi-stage pipelines with external verifiers.
- Strong Empirical Performance:
The method achieves state-of-the-art results on the GenEval benchmark, with a substantial +0.25 improvement over its baseline.
The gains are especially pronounced in difficult compositional tasks (Position: +0.69, Attribution: +0.30), indicating a genuine improvement in model understanding.
- Excellent Content Preservation:
The architectural design successfully enables localized edits while preserving the overall image structure.
This is quantitatively supported by SCoT achieving the highest CLIP-Image similarity score (0.96) among all compared one-step editing baselines.
- Sound and Well-Motivated Architecture:
The design choices, including the unified VAE encoding path to mitigate domain shift and the zero-initialized dual-branch cross-attention for controlled guidance, are technically sound and well-justified for the task.
- Clear Presentation:
The paper is well-written, clearly organized, and the proposed method is explained effectively with helpful diagrams and visual examples.
The training details are transparent, which aids in understanding the experimental setup.

**Weaknesses:**

- Narrow Evaluation Scope:
The paper's claims are almost exclusively validated on the GenEval benchmark.
While the results are strong, this over-reliance makes it difficult to assess generalization.
The evaluation lacks human preference studies, which are crucial for judging the subtle quality of corrections.
- Potential Data Pipeline Bias:
The training data is synthesized using a pipeline that relies on GenEval templates and other powerful models (GPT-4o, FLUX).
This raises concerns about distributional leakage, where the model may be unintentionally fine-tuned on the specific style and structure of the test benchmark, potentially inflating performance.
- Insufficient Efficiency Analysis:
The paper claims efficiency improvements but lacks a rigorous computational cost analysis.
It does not provide essential metrics like wall-clock time or FLOPs-per-image compared to baselines like Reflect-DiT and best-of-N sampling under an identical compute budget.
- Limited Ablation Studies:
The ablations primarily focus on the presence of interaction layers.
The paper would be strengthened by ablations on the data itself, such as the impact of the positive-to-negative pair ratio, the effect of noise in the training data from the detector, and the importance of the zero-initialization strategy.
---
- General Limitations:
The authors should include a dedicated limitations section discussing the following:
    - The model's performance is fundamentally dependent on the quality and biases of the third-party models (object detector, GPT-4o, FLUX) used to create the training data.
    - The current framework is tailored to fix object-centric errors identifiable by detectors and may not generalize to more abstract, stylistic, or subjective flaws.
    - The potential for misuse of such a targeted editing tool for subtle and potentially deceptive content manipulation is not discussed.
    - The reproducibility of the core contribution is hampered by the reliance on a complex, proprietary data generation pipeline.

**Questions:**

- Regarding data provenance, beyond filtering prompts that appear in the test set, what steps were taken to ensure the training data templates are sufficiently distinct from the GenEval test templates to prevent distributional leakage?
Could you provide statistics on the lexical or structural similarity between your training prompts and the test set?
- Could you provide a precise computational cost comparison?
Specifically, please report the per-image wall-clock time and/or FLOPs for generating a final corrected image with SCoT (Best-of-20) versus Reflect-DiT (N=20) and a naive Best-of-20 baseline, using the same hardware.
- How does the model behave under repeated applications?
If a correction is still imperfect, does a second pass with SCoT continue to improve the image, or does performance saturate or even degrade? Showing a k-step (e.g., k=1 to 5) performance curve would be very insightful.
- The data generation pipeline relies on an object detector to identify errors.
What was the estimated error rate of the detector, and how does this label noise in the training data affect the model's final performance? An experiment analyzing the model's robustness to synthetic feedback noise would be valuable.

---

### Official Review · Reviewer_N5PS · 2025-11-01

**Soundness:** 2
**Presentation:** 1
**Contribution:** 2
**Rating:** 4
**Confidence:** 3

**Summary:**

The authors propose SCoT, a method that enables image generators to correct specific erroneous regions in generated images while keeping the rest of the content unchanged.
The method trains on synthetic positive–negative image pairs created from Reflect-DiT outputs and GPT-4o-assisted corrections.
Architecturally, SCoT augments the SANA-1.0-1.6B base model with dual attention branches for image and text conditioning, improving consistency and prompt alignment.

**Strengths:**

Shows clear improvements on GenEval metrics, demonstrating effectiveness in targeted correction.
Conceptually simple --> turns test-time resampling into a self-reflective correction process.
Efficient: adds minimal overhead and runs on modest hardware.

**Weaknesses:**

- Relies on synthetic GPT-4o-generated training data, raising reproducibility concerns.
- Appears to reduce image diversity, potentially limiting generative variability.
- The paper is difficult to follow, with redundant phrasing and unclear explanations (e.g., line 201: “low inference cost, and fast sampling speed”).
- Lacks quantitative evaluation of runtime gains and edit localization metrics.

**Questions:**

- How much does SCoT modify the original image (e.g., pixel-level or feature-level distance)?
- What is the actual speed-up achieved compared to standard inference?
- Is the reliance on synthetic supervision scalable without LLM-generated data?

---

### Official Review · Reviewer_581J · 2025-11-03

**Soundness:** 1
**Presentation:** 2
**Contribution:** 1
**Rating:** 2
**Confidence:** 5

**Summary:**

This paper proposes SCoT for a test-time self-correction framework for text-to-image generation. SCoT aims to use the same model to generate an image from text first and then correct the generated image to improve the quality. The training data of positive and negative samples for self-correction is synthetized using GPT-4o and Flux.1-Kontext. SANA-1.0-1.6B is finetuned on the synthetic training samples after extending its attention layer for image reference. Experiments on the GenEval benchmark show performance gains and qualitative improvements compared with the previous best-of-N method for test-time scaling.

**Strengths:**

S1. This work targets important and practical topics – self-correction of image generative models. The capability of self-correction can incorporate reasoning processes to improve the image generation performance as language models have brought significant performance gains by test-time scaling.

S2. The proposed framework is easy to follow and logically extends prior test-time scaling works toward a self-contained correction mechanism.

S3. Empirical results show outperformance than SFT (best-of-N) and previous method (Reflect-DiT), showing the potential of the proposed approach.

**Weaknesses:**

W1. Limited novelty and implicit capability for self-correction. Although the framework aims to incorporate reasoning capability for self-reasoning, the trained model cannot explicitly reason about the erroneous regions but implicitly understand the errors during image-to-image generation. The proposed architecture is a well-known trick to extend the attention layer of image generation models such as IP-Adapter or ControlNet.

W2. The self-correction capacity is limited in a single-step correction. As a test-time scaling approach, it’s natural to expect to replace best-of-N with a self-correction method. However, this paper still uses best-of-N, while adding one-step image correction to boost the performance. Also, the proposed method lacks multiple-step image correction to conduct test-time scaling only by this self-correction method.

W3. Lack of in-depth analysis. Although the proposed SCoT outperforms best-of-N, there is no in-depth analysis to understand how the proposed method works. Please refer to the questions below.

W4. Literature review can be conducted more. For example, in masked token modeling of image generation, there are some famous approaches to add self-correction of generated images to improve the performance [NewRef-1, NewRef-2].

[NewRef-1] Discrete Predictor-Corrector Diffusion Models for Image Synthesis (Lezama et. al., ICLR’23).
[NewRef-2] Draft-and-Revise: Effective Image Generation with Contextual RQ-Transformer (Lee et. al., NeurIPS’22).

**Questions:**

In addition to the weaknesses above, I think resolving my concerns below can significantly help improve this paper.

Q1. Why is self-correction conducted only in a single step? What if the model conducts self-correction multiple times? I first thought that the proposed method replaces the best-of-N method with self-correction for effective test-time scaling, but the experiments still use the best-of-N method. Cannot the self-correction method replace the best-of-N?

Q2. In Table 1, how is the best-of-N method conducted? Which reward function is used to select the best sample for each category?

Q3. Can the authors provide more evaluation measures? Especially, if GenEval score is used for best-of-N, adding a new evaluation metric will help comparing generalization performance, considering the evaluation of image generation always requires a multi-folded approach.

Q4. In Table 1, considering that the best-of-20 actually conducts 40 inferences, comparing SFT (best-of-20) with SCoT (best-of-10) is fair. In addition, comparing the performance with various N in best-of-N can also help understanding the efficacy of the proposed self-correction method.

Q5. I’m wondering about the performance of the simple combination of GPT-4o and Flux 1 Kontext, which are used for training data generation. Can the fine-tuned SANA outperform the training data generation framework?

Q6. Can more benchmarks be used for performance comparison? For example, T2I-CompBench can also be used for comparing the performance.

---

### Note · Authors · 2025-12-01

I have read and agree with the venue's withdrawal policy on behalf of myself and my co-authors.